# Accelerating Large Language Model Inference via Speculative Decoding with Progressive Tree Drafting

## Abstract

The draft-then-verify decoding paradigm, introduced by speculative decoding methods, has demonstrated remarkable performance in alleviating the memory-bound bottleneck and accelerating the inference speed of Large Language Models (LLMs) while maintaining the quality of generated content. Recent studies show that the intrinsic robustness of LLMs can be exploited in a training-free and architecture-agnostic manner, suggesting that auxiliary models or structural modifications are not strictly necessary for draft generation. However, existing methods fail to fully leverage this robustness, leading to substantial redundant and repeated computations. Building on this insight, we propose Progressive Tree Drafting (PTD), a new inference acceleration strategy that further extends this line of work. PTD organizes the drafting process into a progressively updated tree structure, where controlled perturbations are injected to guide generation and a stepwise pruning mechanism enabling the model to produce coherent yet diverse drafts at manageable computational cost. By efficiently coordinating the drafting and verification stages, PTD achieves up to $2\times$ decoding speedup across different open-source models and benchmarks. Our code is available at https://anonymous.4open.science/r/PTD-D354.

## 1 Introduction

Large language models(LLMs) leverage parallel training on extensive datasets to enhance both training efficiency and text generation capabilities. However, during the autoregressive decoding process, tokens are generated sequentially, requiring all model parameters to be loaded into the on-chip buffer at each decoding step. As a result, the inference process often becomes constrained by GPU bandwidth, putting the inference system in a memory-bound state.

To address this issue, speculative decoding transforms the token-by-token decoding strategy into a candidate parallel verification process, introducing a new draft-then-verify decoding paradigm. The key to speculative decoding lies in obtaining high-quality drafts. A common approach to generating drafts is to employ smaller and faster draft models that condition on the current context (Xia et al., 2023; Leviathan et al., 2023; Chen et al., 2023; Miao et al., 2024; Yang et al., 2024). These methods often incur significant communication overhead and require substantial training effort to obtain and align the draft models. To mitigate this issue, some methods retrieve drafts from a pre-constructed corpus (He et al., 2023; Yang et al., 2023) but often face a trade-off between contextual relevance and generality. An alternative line of work leverages the target LLM itself to produce drafts. Some works attempt to sample features from intermediate layers of the target model (Li et al., 2024a;b) or modify the output layer (Cai et al., 2024; Stern et al., 2018; Li et al., 2025), enabling the LLM to generate multiple drafts or decode multiple tokens within a single forward pass. However, these methods require modifications to the model architecture and parameters, and often involve additional training for the modifications.

In fact, owing to their large parameter scale and extensive training data, LLMs often maintain semantic coherence even when the input is perturbed (Zhu et al., 2024; Gao et al., 2025). This phenomenon offer us an opportunity to break the strict sequential dependency of autoregressive decoding. Self-Draft (Gao et al., 2025) takes an initial step in this direction by adopting a multi-branch drafting

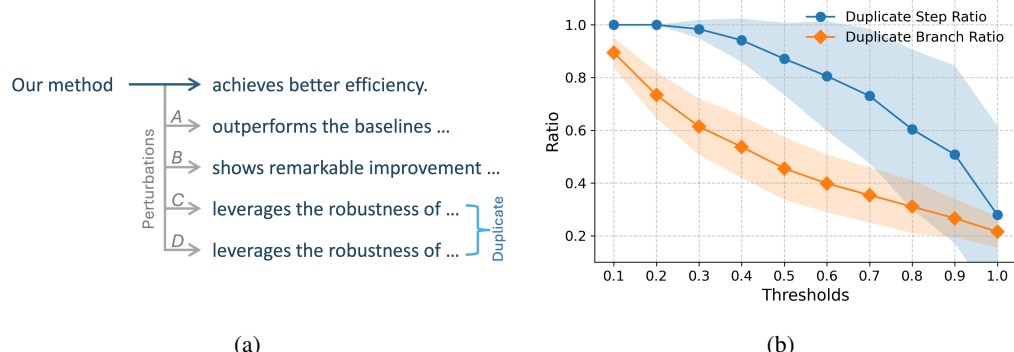

(a)  (b)

Figure 1: A drafting example (a) and the branch similarity analysis (b) of Self-Draft (Gao et al., 2025). Blue: proportion of steps with at least two branches above the similarity threshold; Orange: overall proportion of such branches.

strategy that leverages LLM robustness to generate candidate drafts. Figure 1a presents a typical drafting process of the Self-Draft, where different perturbations are introduced across branches to promote diversity. However, our analysis of branch diversity reveals that the method still suffers from excessively high similarity among branches. As shown in Figure 1b, more than half of the decoding steps contain branches with over 80% similarity, leading to a substantial waste of computational resources.

To address this issue, we propose **Progressive Tree Drafting (PTD)**, a novel training-free and model-agnostic speculative decoding strategy. This approach modifies the decoding process by introducing an additional drafting task guided by a progressive tree structure as an input perturbation, thereby better leveraging the robustness of LLMs and eliminating the resource waste caused by redundant drafting. Through the corresponding expansion and stepwise pruning algorithms, the tree structure supports incremental expansion, prefix sharing, and adaptive pruning. These capabilities not only enable coherent and diverse generation, but also facilitate computational reuse across decoding steps while keeping the additional overhead effectively controlled. Unlike other speculative decoding methods, our approach requires neither auxiliary small models nor architectural modifications, making it readily applicable to most autoregressive LLMs.

The main contributions of this paper are summarized as follows:

- First, we identify a key bottleneck in existing linear branch perturbation–based methods: the draft module fails to ensure sufficient diversity, leading to substantial waste of computational resources.

- Second, we introduce the progressive tree drafting strategy, which perturbs the input progressively to generate diverse and coherent drafts, fully exploiting the robustness of LLMs while significantly reducing redundancy computation.

- Finally, experimental results demonstrate that our method outperforms state-of-the-art approaches across various models and benchmarks, exhibiting strong acceleration capabilities and adaptation.

The structure of this paper is as follows: First, we review related works, followed by a detailed description of the proposed method. Then, we present experimental results to validate its effectiveness. Finally, we conclude the paper and discuss potential directions for future research.

## 2 RELATED WORKS

Speculative decoding has emerged as a promising approach for accelerating autoregressive generation in LLMs without compromising output quality. The core idea is to generate candidate tokens and then verify these candidates using the full target LLM. This paradigm was first formalized by Xia et al. (2023), and later extended in various directions. For instance, Leviathan et al. (2023) ap-

plied speculative decoding to Transformer-based models, demonstrating substantial speedups. Chen et al. (2023) further refined this approach by introducing speculative sampling, which adds stochasticity to candidate generation to increase diversity.

Subsequent efforts further optimized this paradigm. Recent works have extended the speculative decoding paradigm to improve draft quality and verification efficiency. MCSD (Yang et al., 2024) proposes decoding multiple candidate tokens at each step using a draft model, which are then verified in parallel by the target model to increase acceptance rates. Ouroboros (Zhao et al., 2024) introduces an additional candidate pool as a warm start to enhance the efficiency of the drafting model in generating multiple candidate drafts. SpecInfer (Miao et al., 2024) further explores using multiple draft models to generate diverse candidate sequences, which are merged before verification, also aiming to boost acceptance.

The EAGLE series (Li et al., 2024a;b) departs from using traditional autoregressive draft models and instead trains a prediction model that generates multiple future tokens conditioned on the hidden states of the target model. This greatly reduces drafting time while maintaining semantic relevance. In Judge Decoding (Bachmann et al., 2025), the authors introduce a learned judge head to relax the strict alignment constraint during verification, allowing the target model to accept drafts that are not fully aligned but still coherent. To further reduce the drafting cost and communication overhead between models. REST (He et al., 2023) and LLMA (Yang et al., 2023) replace the drafting model with a pre-constructed draft corpus, enabling retrieval-based draft generation with lower latency.

Furthermore, some works attempt to obtain drafts without relying on additional draft models or external corpora to further enhance the applicability of inference acceleration methods. Draft&Verify (Zhang et al., 2023), LayerSkip (Elhoushi et al., 2024), and Kangaroo (Liu et al., 2024) perform draft generation directly within the target LLM itself, leveraging intermediate layer embeddings to train predictors for future tokens. Medusa (Cai et al., 2024) and Blockwise Decoding (Stern et al., 2018) introduce additional output heads, each responsible for predicting several future positions in parallel.

Some works go even further by modifying the decoding process to obtain drafts without any additional training. LADE (Fu et al., 2024) adapts the Jacobi decoding algorithm for autoregressive models, achieving inference acceleration without requiring external assistance or extra training. Self-Draft (Gao et al., 2025) leverages the robustness of LLMs by multiple linear branches to extract drafts, which are then validated for correctness, reducing reliance on separate draft models.

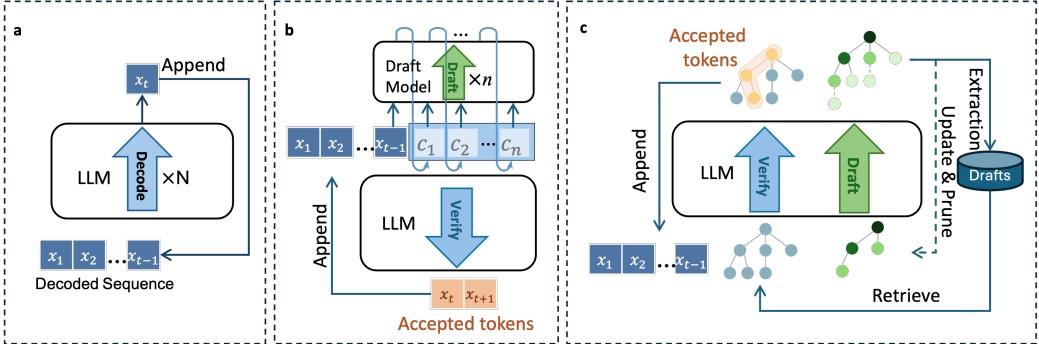

Figure 2: An overview of autoregressive decoding (a), vanilla speculative decoding (b), and PTD (c).

## 3 METHOD

### 3.1 OVERVIEW OF TREE-DRAFT

Figure 2 presents a comparison of the progressive tree drafting inference method against autoregressive decoding and vanilla speculative decoding. The conventional autoregressive decoding (Figure 2.a) process operates as follows. Given an input prompt consisting of $t-1$ tokens, denoted as $\mathbf{X} = [x_1, x_2, \cdots, x_{t-1}]$, the LLM computes the probability distribution of the next token, repre-

sented as $P(y_t|\mathbf{X})$. The next token $x_t$ is then sampled from this distribution using a strategy $\mathcal{S}$, such as greedy decoding, Top-K, or Top-P sampling. Once generated, $x_t$ is appended to $\mathbf{X}$, forming an updated input sequence, and the process iterates until completion. This decoding mechanism can be formally expressed as:

$$x_t = \mathcal{S}(P(y_t|\mathbf{X}))$$

In speculative decoding (Figure 2.b), the majority of the autoregressive generation is handled by a lightweight draft model, which reduces communication overhead, while the target model is responsible for verifying the outputs in parallel. Our approach (Figure 2.c) reformulates the conventional autoregressive decoding process into a progressive tree drafting and candidate tree verification process. In the following content of this section, we will elaborate on each of these processes in detail.

## 3.2 PROGRESSIVE TREE DRAFTING

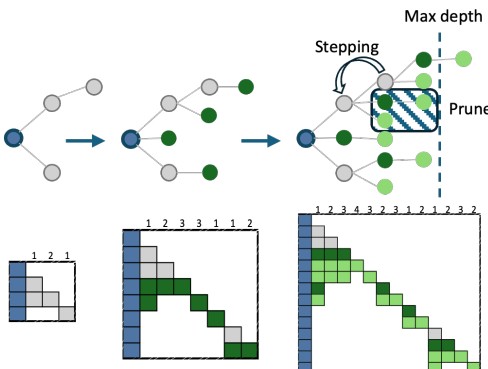

Figure 3: Illustration of progressive tree drafting and its corresponding attention mask, which is determined by the deep first traversal. The nodes in the upper part of the figure represent tokens, and the edges between nodes indicate the partial order relations among tokens. The numbers above the attention mask matrix are the relative positions.

The progressive tree drafting decoding strategy leverages the context-relative and potentially usable parse generated by the LLM under perturbation. Figure 3 shows an illustration of the initialization and progressive updating of the progressive drafting tree. In the diagram, the blue nodes represent the decoded tokens, while the gray nodes represent randomly initialized perturbation tokens, which are used to initialize the progressive draft tree to ensure the draft content diversity as it grows. The gradient green nodes illustrate the incremental expansion of the draft tree, and the connecting lines between nodes denote partial order relationships, which represent the receptive field of each draft token. We next provide a detailed description of the draft tree expansion process and the extraction of draft sequences from it.

In decoder-only Transformer architectures, the attention mask is typically implemented as a lower triangular matrix, ensuring that each token can only attend to preceding tokens in the sequence. To enable large language models to reason correctly over more complex data structures, such as the tree-structured inputs introduced in this work, the attention mask must be adapted accordingly.

Specifically, given a randomly initialized shallow tree $T^0 = (V^0, E^0)$, which $V^0$ means the random initialized nodes and $E^0$ are the edges between them. To ensure semantic consistency, each node in the tree should be only aware of tokens that precede it along its branch and should remain unaware of tokens from other branches. Formally, for each node $v$ in the draft tree, we can determine its aware nodes $\pi(v)$ as:

$$\pi(v) = \{v\} \cup \pi(\mathcal{P}(v)),$$

where $\mathcal{P}(v) = \{u \in V | (u, v) \in E\}$, stand for the parent node of the node $v$. We can also determine the positional encoding of node v based on $\pi(v)$.

Thus, the inference on the draft tree can be formulated as:

$$x_t, \mathcal{D} = \mathcal{S}\left(P\left(y_t, \mathbf{y}_T | [\mathbf{X}; T^{i-1}]\right)\right),$$

where $\mathcal{D}$ means all draft tokens that are generated by the draft tree $T^{i-1}$. The main experiments in this paper adopt a greedy strategy to obtain these tokens, and we also examine alternative approaches in experiments. Then we can obtain the $i$-th progressive drafting tree $T^i$ by expanding the previous iteration draft tree $T^{i-1}$ based on $\mathcal{D}$, that is:

$$V^i = V^{i-1} \cup \{d_v | \forall v \in V^{i-1}\}$$

$$E^i = E^{i-1} \cup \{(v, d_v) | \forall v \in V^{i-1}\}$$

where $d_v \in \mathcal{D}$ is the draft token of node $v$ under the current context with prefix $\pi_v$.

Generally, the number of nodes in the draft tree ensures the diversity of the drafts it generates, and the expansion process maintains the semantic coherence between the adjacent nodes in the tree. However, the computational overhead introduced by the draft tree increases progressively as it grows. Hence, it is necessary to impose constraints on its growth to prevent excessive size, which could otherwise degrade the overall decoding speed.

Specifically, we constrain the size of the draft tree along two dimensions: width and depth. For width, we limit the number of child nodes per node to prevent low-confidence draft tokens from frequently altering the tree structure, which could compromise the overall quality and coherence of the generated content by the draft tree.

For depth, we adopt a stepping mechanism to regulate the expansion of the tree. As illustrated in Figure 3, when the immediate sub-tree $T_s$ of the root $r$ exceeds a predefined depth threshold, only the earliest-added child node and its descendants are retained. This retained branch is then treated as the new sub-tree replacing $T_s$, while all other branches are pruned. This inheritance mechanism helps preserve the semantic coherence and contextual relevance of the draft tree.

Through progressive updates and a stepping mechanism, the draft tree enables the extraction of draft content based on the current context and perturbations of the draft tree. Specifically, any subtree $T'$ in the draft tree $T^i$ will be merged with the cached candidate tree $T$ that shares the same root node value. We define the following recursive merging function $\mathcal{M}$ for any two trees $T$ and $T'$ with same root $r$:

$$\mathcal{M}(T, T') = \begin{cases} (V \cup v, E \cup (r, v)), \forall v \in \sigma(T') - \sigma(T) \\ \mathcal{M}(T_v, T'_v), \forall v \in \sigma(T') \cap \sigma(T) \end{cases},$$

where $\sigma(T)$ means the direct child nodes of the root $r$ of tree $T$, and $T_v$ is the subtree with root of $v$ in tree $T$.

### 3.3 CANDIDATE VERIFICATION

Alongside the autoregressive decoding process and the drafting process, a candidate tree validation process is concurrently executed during the forward pass. Given the partially decoded token sequence $\mathbf{X}$, we retrieve corresponding drafts from the draft pool, forming the candidate tree $\mathcal{C}_\mathbf{X}$.

To verify this candidate tree, we apply the same attention mask and positional encoding strategy as used in the drafting process. Consequently, after a forward pass through the LLM, each node in $\mathcal{C}_\mathbf{X}$ produces a verification token conditioned on its prefix. Together with the autoregressive decoding process and the progressive tree drafting process, we formulate the overall model forward process as follows:

$$x_t, \mathcal{D}, \mathcal{V} = \mathcal{S}\left(P(y_t, \mathbf{y}_T, \mathbf{y}_\mathcal{C} | [\mathbf{X}; T^{i-1}; \mathcal{C}_\mathbf{X}])\right),$$

where $\mathcal{V}$ denotes the verification tokens that are generated by each node and its prefix in the candidate tree.

Finally, the accepted tokens $\mathbf{X}'$ can be obtained by identifying all eligible edges $\mathcal{E}$ in the candidate tree. Under the **greedy decoding strategy**, the verification tokens $\mathcal{V}$ for all nodes $V_{\mathcal{C}_\mathbf{X}}$ of the candidate tree $\mathcal{C}_\mathbf{X}$ are selected based on the model's highest-probability predictions. Eligible edges are identified recursively by verifying whether a node's verification token appears among its child nodes. That is:

$$\mathcal{E} = \{(n, \mathcal{V}_n) | \mathcal{V}_n \in \sigma(n), \forall n \in V_{\mathcal{C}_\mathbf{X}}\}.$$

For the **sampling decoding strategy**, we determine whether each token is accepted using a without-replacement sampling method based on normalized probabilities, following an approach similar to

LADE (Fu et al., 2024) and SpecInfer (Miao et al., 2024). Specifically, starting from the root node of the candidate draft tree, the LLM produces a probability distribution $P_v$ over the next token at each node $v$. Each node may have multiple successor nodes $[c_1, c_2, ..., c_k]$, and a sampling process is iteratively applied to these $k$ candidates.

At each iteration, a random number $r \sim \mathcal{U}(0, 1)$ is drawn, and the candidates are traversed in order. If $r \leq P_i$, the candidate $c_i$ is selected, and the edge between $c_i$ and its parent node is marked as eligible and appended to the eligible edge set $\mathcal{E}$. If not, $P_i$ is set to zero, and the remaining probabilities are renormalized. This process continues until a candidate satisfies $r \leq P_i$, ensuring that the final selection remains faithful to the original distribution. We provide the PTD decoding strategy algorithm and the formal proof of the consistency of the decoding distribution under this candidate tree sampling strategy in Appendices A, B and C.

The final accepted sequence is the path formed by eligible edges starting from the root node $n_0$. That is,

$$\mathbf{X}' = (n_0, n_1, ...n_k, \mathcal{V}_{n_k})$$

where $\forall i < k, (n_i, n_{i+1}) \in \mathcal{E}$ and $\mathbf{X}'$ are the tokens we decoded in a single model forward pass.

## 4 EXPERIMENTS

### 4.1 SETTINGS

**Benchmarks.** We selected various benchmark datasets to evaluate the performance of our decoding method across different scenarios. First, we used MT-Bench (Zheng et al., 2023) to assess the overall effectiveness of our approach. This benchmark comprises eight distinct types of tasks, each comprising 10 test problems. Additionally, we randomly sampled 100 questions from the GSM-8k (Cobbe et al., 2021) dataset to evaluate our method's performance in mathematical problem-solving tasks. For code completion evaluation, we sampled 100 problems from the test set of the MBPP (Austin et al., 2021) dataset and used the entire HumanEval (Chen et al., 2021) dataset.

**Baselines.** We adopt the autoregressive decoding (AR) method, the speculative decoding (SpeDe) (Leviathan et al., 2023) method (with the draft model of LLaMA-68M (Miao et al., 2024)). Lookahead Decoding (LADE) (Fu et al., 2024) and Self-Draft (Gao et al., 2025) (without pre-built cache) method, which requires neither an auxiliary model nor additional training, as our baselines, all parameters of this method are set with default values.

**Models.** We selected the LLaMA2-7B/13B-Chat (L-7B/13B) and Qwen2.5-7B/14B/32B-Instruct (Q-7B/14B/32B) models for general generation tasks (MT-Bench) and mathematical reasoning (GSM-100), while CodeLLaMA-7B/13B-Instruct (CL-7B/13B) models were used for code generation tasks (HumanEval, MBPP-100).

**Metrics.** We evaluate decoding strategies using five metrics: throughput (TP), decoding efficiency (DE), hit rate (HR), accept length (AL), and computational overhead. TP measures tokens generated per second. Computational overhead is the average extra tokens decoded per step during drafting (Dft) and verification (Ver). DE measures generated tokens per forward pass, influenced by HR and AL, which reflect the diversity and coherence of the draft, respectively. The relationship among DE, AL, and HR can be expressed by the following equation:

$$\mathrm{DE} = \mathrm{HR} \cdot \mathrm{AL} + (1 - \mathrm{HR}).$$

All experiments were conducted on NVIDIA L20 GPUs (48 GB RAM) using BF16 precision to enhance computational efficiency. Inference was performed consistently with a batch size of one throughout. Unless otherwise specified, all draft tokens are obtained using the greedy method.

### 4.2 RESULTS

#### 4.2.1 MAIN RESULTS

Table 1 presents the throughput improvements achieved by different methods on different benchmarks under the sampling decoding strategy, and we also provided the greedy decoding strategy in Appendix E. We set the maximum depth of the draft tree to 6 and the maximum number of children

| Benchmark | Model | AR | SpeDe | | LADE | | Self-Draft | | PTD | |
|---|---|---|---|---|---|---|---|---|---|---|
| | | TP(Std) | TP(Std) | Imp. | TP(Std) | Imp. | TP(Std) | Imp. | TP(Std) | Imp. |
| MT-Bench | L-7B | $39_{\pm3.9}$ | $52_{\pm7.0}$ | 32% | $58_{\pm9.8}$ | 49% | $60_{\pm12.1}$ | 54% | $\mathbf{65}_{\pm11.1}$ | **66%** |
| | L-13B | $24_{\pm1.7}$ | $36_{\pm5.8}$ | 39% | $33_{\pm4.9}$ | 39% | $37_{\pm6.6}$ | 54% | $\mathbf{40}_{\pm5.8}$ | **65%** |
| | Q-7B | $36_{\pm4.3}$ | \ | \ | $51_{\pm8.8}$ | 43% | $53_{\pm12.5}$ | 48% | $\mathbf{61}_{\pm15.4}$ | **71%** |
| | Q-14B | $20_{\pm1.9}$ | \ | \ | $29_{\pm5.0}$ | 45% | $30_{\pm6.6}$ | 52% | $\mathbf{35}_{\pm7.6}$ | **76%** |
| | Q-32B | $10_{\pm0.6}$ | \ | \ | $16_{\pm3.0}$ | 56% | $16_{\pm3.3}$ | 63% | $\mathbf{19}_{\pm3.9}$ | **86%** |
| GSM-100 | L-7B | $43_{\pm0.9}$ | $58_{\pm4.7}$ | 35% | $73_{\pm5.6}$ | 68% | $74_{\pm6.2}$ | 70% | $\mathbf{82}_{\pm7.5}$ | **88%** |
| | L-13B | $26_{\pm0.4}$ | $36_{\pm2.9}$ | 39% | $41_{\pm3.3}$ | 59% | $44_{\pm4.5}$ | 72% | $\mathbf{48}_{\pm4.2}$ | **86%** |
| | Q-7B | $39_{\pm1.9}$ | \ | \ | $61_{\pm6.3}$ | 55% | $61_{\pm7.4}$ | 56% | $\mathbf{73}_{\pm13.3}$ | **87%** |
| | Q-14B | $22_{\pm0.5}$ | \ | \ | $34_{\pm3.8}$ | 59% | $35_{\pm4.2}$ | 63% | $\mathbf{41}_{\pm5.9}$ | **92%** |
| | Q-32B | $10_{\pm0.2}$ | \ | \ | $18_{\pm1.5}$ | 76% | $19_{\pm1.6}$ | 84% | $\mathbf{23}_{\pm2.1}$ | **118%** |
| HumanEval | CL-7B | $42_{\pm1.6}$ | \ | \ | $61_{\pm5.7}$ | 45% | $68_{\pm7.4}$ | 61% | $\mathbf{71}_{\pm7.4}$ | **70%** |
| | CL-13B | $25_{\pm0.7}$ | \ | \ | $36_{\pm4.6}$ | 45% | $42_{\pm5.4}$ | 68% | $\mathbf{43}_{\pm5.6}$ | **73%** |
| MBPP-100 | CL-7B | $44_{\pm0.8}$ | \ | \ | $75_{\pm7.5}$ | 70% | $82_{\pm6.7}$ | 87% | $\mathbf{90}_{\pm8.9}$ | **105%** |
| | CL-13B | $26_{\pm0.3}$ | \ | \ | $43_{\pm4.1}$ | 65% | $49_{\pm4.3}$ | 90% | $\mathbf{54}_{\pm5.4}$ | **107%** |

Table 1: Throughput and Improvement (Imp.) under sample decoding(temperature=0.5) for PTD, Auto-Regressive decoding (AR), the vanilla Speculative Decoding (SpeDe) using a LLaMA-68M (Miao et al., 2024) draft model, Lookahead decoding (LADE) (Fu et al., 2024), and Self-Draft (Gao et al., 2025).

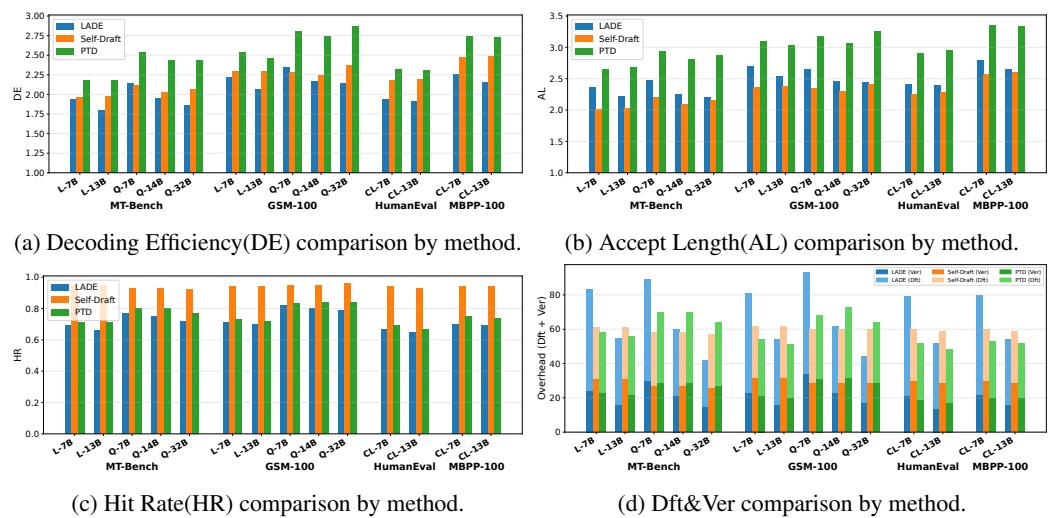

(a) Decoding Efficiency(DE) comparison by method.

(b) Accept Length(AL) comparison by method.

(c) Hit Rate(HR) comparison by method.

(d) Dft&Ver comparison by method.

Figure 4: Draft content quality analysis.

per node to 4, which will be further discussed in the next section. As can be observed, compared to existing inference acceleration techniques, our proposed PTD method consistently achieves more significant speedups across various tasks and models. Notably, our method achieves more substantial improvements on the mathematical reasoning benchmark, GSM-100, and the Python coding benchmark, MBPP-100. This is because the tasks in both benchmarks are well-defined, and the search space for generation is relatively small. As a result, our drafting strategy can produce coherent drafts with high coverage, leading to significant acceleration in reasoning.

Figure 4 provides a more in-depth analysis of decoding efficiency(DE), hit rate(HR), candidate draft acceptance length(AL), and overhead(Dft/Ver), which reveals the underlying causes of speed differences among the methods. The DE metric reflects overall decoding efficiency, consisting of two components: Accept Length and Hit Rate. On this metric, PTD demonstrates a comprehensive advantage. More specifically, although our method lags slightly behind Self-Draft in Hit Rate—mainly

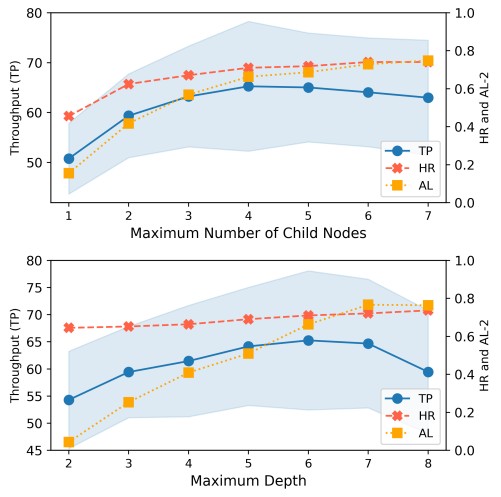 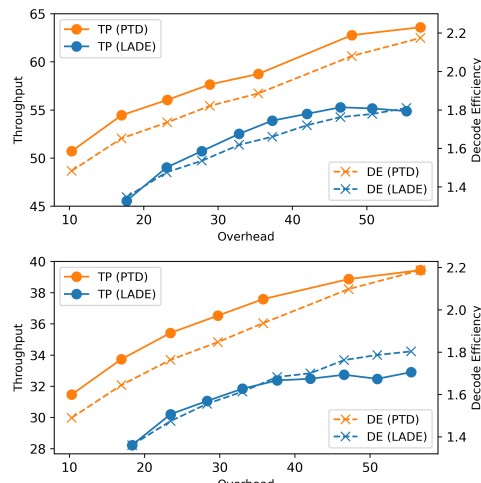

Figure 5: The impact of the maximum number of child nodes (top) and maximum depth (bottom) of the draft tree on throughput, hit rate, and accept length (offset by -2 for visualization) of LLaMA-7b on MT-Bench.

Figure 6: Draft efficiency for LLaMA-7b(top) and LLaMA-13b(bottom) LADE and PTD on MT-bench.

because Self-Draft leverages additional external corpora to improve its hit probability—it achieves a significant lead in Accept Length. This indicates that the drafts generated by PTD are of higher quality and exhibit much stronger contextual coherence compared to those from Self-Draft. At the same time, the additional overhead of PTD is comparable to that of Self-Draft and generally superior to LADE, implying that PTD does not introduce noticeable forward latency during model inference.

### 4.2.2 DRAFT TREE ANALYSIS

A key factor influencing the performance of our approach is the overhead introduced by the drafting tree. To control its complexity, we constrain the number of child nodes and the overall depth of the tree. In this section, we analyze how these two parameters affect the acceleration performance and determine their optimal configuration.

We first evaluate the impact of the maximum number of child nodes on the acceleration performance. In this experiment, the maximum depth of the drafting tree is fixed at 6, and the maximum number of child nodes per node is varied. As shown in the top part of Figure 5, increasing the child node limit initially leads to improvements in both draft hit rate and accepted length, which then plateau. Meanwhile, the overall decoding speed increases at first but eventually decreases. This is because the benefit gained from a larger drafting tree can no longer offset the additional computational overhead it introduces, resulting in a decline in overall decoding throughput.

Based on the previous results, we analyze the impact of varying the drafting tree depth by fixing the maximum number of child nodes per node at 4 and gradually increasing the tree depth limit. As shown in the bottom part of Figure 5, throughput also exhibits a rise-then-fall trend due to the overhead of deeper trees. The accepted length, however, shows a steady increase. This suggests that increasing the tree depth moderately can improve the coherence of generated drafts, thereby allowing longer segments to be accepted when a draft is successfully matched. In contrast, the hit rate remains relatively stable, with no clear trend, indicating that the depth of the tree has no significant correlation with the hit rate.

Overall, the experimental results show that our method demonstrates a significant improvement in decoding acceleration across a wide range of tree depth and child number settings, exhibiting strong robustness. Based on the results, in this paper, unless otherwise specified, we set the child number to 4 and the maximum tree depth to 6.

### 4.2.3 DRAFT EFFICIENCY

Additionally, we analyzed the drafting efficiency of our method, especially comparing its inference acceleration performance with LADE under the same computational overhead introduced by the drafting and verification phases. As shown in Figure 6, we can observe that our method outperforms LADE overall, achieving a decoding speed improvement comparable to that of LADE with only about half of the additional computational overhead for the 7B model, and only one-third of the additional overhead for the 13B models, respectively. Even with little additional overhead, our approach achieves significant acceleration, demonstrating the efficiency and advantages of our drafting method compared to LADE in large-scale inference services.

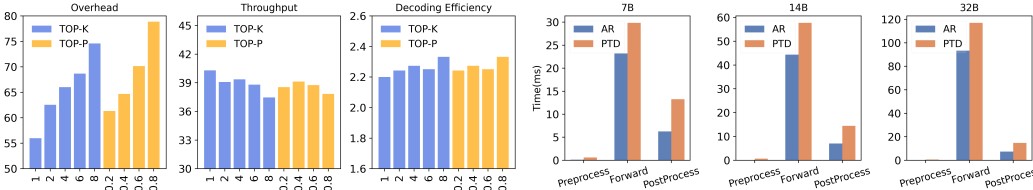

Figure 7: Sample-based tree updating method.   Figure 8: Run time analysis for Qwen models.

### 4.2.4 SAMPLE STRATEGY FOR THE DRAFT TREE EXPANSION

We also analyzed the impact of draft tree expansion under different sampling strategies. Specifically, in addition to the previously discussed greedy method, we further analyzed the top-k and top-p sampling methods. We first obtain the top-k or top-p distribution for each node, then we sample draft tokens according to their corresponding probabilities and extend the draft tree. Figure 7 shows the results of LLaMA-13b on MT-Bench. Regarding the sampling strategy, we observed that the overhead is larger than that of the greedy strategy (top-k with k = 1) on average. This is because under these decoding strategies, the draft tree tends to exhibit greater diversity and uncertainty, which leads to faster tree growth. However, the increased overhead did not bring significant improvements in decoding throughput and efficiency.

### 4.2.5 OVERHEAD ANALYSIS

Compared to autoregressive decoding, PTD introduces the following additional overhead. Before the model's forward pass, we need to retrieve the candidate tree. During the forward pass, we perform parallel inference to generate additional draft tokens and verification results. After the forward pass, the candidate pool and draft tree need to be updated. Figure 8 shows the comparison between autoregressive decoding and PTD with different model sizes. Compared to autoregressive methods, the PTD method incurs the most significant additional computational overhead in the model's forward pass. This is because we need to perform extra inference on both the draft tree and the candidate tree, but the overhead of updating and retrieving is negligible.

## 5 CONCLUSION

In this paper, we introduce the Progressive Tree Drafting method for LLM inference acceleration. By incorporating incremental expansion and stepwise pruning mechanisms, our approach ensures both the coherence and diversity of drafts while effectively controlling additional overhead, thereby significantly improving overall inference speed. There are also several avenues for further optimization. First, although tree-based perturbations can reduce overhead to some extent via prefix sharing, there exist more dense and semantically structured perturbation methods—such as semantic graphs—that we could leverage in the future for more guided and efficient draft generation. Furthermore, the draft generation process proposed in this paper remains tightly coupled with the decoding process, which may incur excessive overhead when dealing with long texts. Therefore, future work will focus on decoupling these two processes to enhance overall system efficiency.

ETHICS STATEMENT

This work focuses on algorithmic improvements to the efficiency of large language model inference through speculative decoding. Our study does not involve human subjects, private or sensitive data, or the release of new datasets. All experiments are conducted on publicly available, widely adopted open-source models and benchmarks, ensuring reproducibility and transparency.

LLM USAGE STATEMENT

Large language models (LLMs) were used in this work solely as a writing assistance tool, specifically to refine the fluency and clarity of the manuscript text. They were not involved in research ideation, methodology design, data analysis, experimental execution, or result interpretation. All technical contributions, conceptual developments, and scientific claims are entirely the work of the authors. The authors take full responsibility for the final content of the paper.

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

# A   PROGRESSIVE TREE DRAFTING DECODING ALGORITHM

---

**Algorithm 1** Progressive Tree Drafting Decoding Algorithm

---

1: **Input:** Prompt $\mathbf{X} = [x_1, x_2, ..., x_{t-1}]$; max tree depth $d_{\max}$; initial tree $T^0 = (V^0, E^0)$; max length $N$
2: **while** True **do**
3:    {Step 1: Retrieve Candidate Tree}
4:    Retrieve candidate tree $\mathcal{C}_{\mathbf{X}} \leftarrow$ RETRIEVECANDIDATETREE($\mathbf{X}$)
5:    {Step 2: Generate Next Token(s) with Structural Guidance}
6:    $x_t, \mathcal{D}, \mathcal{V} \leftarrow \mathcal{S}\left(P(y_t, \mathbf{y}_T, \mathbf{y}_C \mid [\mathbf{X}; T^{i-1}; \mathcal{C}_{\mathbf{X}}])\right)$
7:    {Step 3: Expand the Draft Tree}
8:    $V^i \leftarrow V^{i-1} \cup \{d_v \mid \forall v \in V^{i-1}\}$
9:    $E^i \leftarrow E^{i-1} \cup \{(v, d_v) \mid \forall v \in V^{i-1}\}$
10:   **if** depth($T^i$) $> d_{\max}$ **then**
11:      $T^i \leftarrow$ STEPANDPRUNE($T^i$)
12:   **end if**
13:   {Step 4: Merge Subtrees into Candidate Pool}
14:   **for** each subtree $T'_s$ in $T^i$ **do**
15:      Update candidate pool by merging trees using $\mathcal{M}$
16:   **end for**
17:   {Step 5: Obtain Eligible Edges}
18:   **if** Using Greedy Decoding **then**
19:      $\mathcal{V} \leftarrow \arg\max(P_{\mathcal{C}})$
20:      $\mathcal{E} \leftarrow \{(n, \mathcal{V}_n) \mid \mathcal{V}_n \in \sigma(n), \forall n \in V^i\}$
21:   **else if** Using Sampling Decoding **then**
22:      $\mathcal{E}, \mathcal{V}_{n_k} \leftarrow$ CANDIDATETREERECURSIVESAMPLE($\mathcal{C}_{\mathbf{X}}$)
23:   **end if**
24:   {Step 6: Append Chosen Path}
25:   $\mathbf{X}' \leftarrow (n_0, n_1, ..., n_k, \mathcal{V}_{n_k})$ s.t. $\forall i < k, (n_i, n_{i+1}) \in \mathcal{E}$
26:   Append $\mathbf{X}'$ to $\mathbf{X}$
27:   **if** $|\mathbf{X}| > N$ **then**
28:      **break**
29:   **end if**
30:   $i \leftarrow i + 1$
31: **end while**
32: **Output:** Generated sequence $\mathbf{X}$

---

# B Candidate Tree Recursive Sampling Algorithm

---

**Algorithm 2** Candidate Tree Recursive Sampling

---

1: **Input:** A node $v$
2: **Output:** Obtain global eligible edges $\mathcal{E}$
3: $C \leftarrow \sigma(v)$ {Children of $v$}
4: **while** $C$ is not empty **do**
5:    **for all** $n \in C$ **do**
6:       Sample $r \sim \mathcal{U}(0, 1)$
7:       **if** $r < P(n)$ **then**
8:          Append $(v, n)$ to $\mathcal{E}$
9:          **call** $\mathcal{V}_{n_k} \leftarrow$ TRAVERSAL$(n)$
10:        **return** $n_k$
11:       **else**
12:          $P[n] \leftarrow 0$
13:          Renormalize $P$ over remaining nodes in $C$
14:       **end if**
15:    **end for**
16: **end while**
17: {If no child selected, sampling based on current node distribution}
18: **return** $\mathcal{S}\left(P(v)\right)$

---

# C    Proof of Distributional Consistency of the Candidate Tree Recursive Sampling Algorithm

We aim to prove that the sampling algorithm described in Appendix A selects each candidate node $n_i$ with probability equal to its original probability $P_i$.

**Sampling Procedure.**    Given a set of candidate nodes $\{n_1, n_2, \ldots, n_k\}$ and associated probabilities $P_i$, the algorithm iteratively samples a random variable $r \sim \mathcal{U}(0, 1)$ and accepts the first node $n_i$ such that $r < P_i$ (after re-normalization, if any earlier nodes have been rejected). If $n_i$ is not accepted, its probability is set to 0, and the remaining probabilities are re-normalized.

**Objective.**    Let $\mathcal{A}_i$ denote the event that node $n_i$ is selected. We aim to prove:

$$\mathbb{P}(\mathcal{A}_i) = P_i, \quad \forall i \in \{1, 2, \ldots, k\}.$$

**Base Case** ($i = 1$).    Node $n_1$ is the first candidate considered. Since no re-normalization has occurred yet, its acceptance probability is:

$$\mathbb{P}(\mathcal{A}_1) = \mathbb{P}(r < P_1) = P_1.$$

**Inductive Step.**    Suppose that for each $j < i$, the probability of selecting node $n_j$ is exactly $P_j$, and the algorithm correctly rejects $n_1$ through $n_{i-1}$ with total probability $R_{i-1} = \sum_{j=1}^{i-1} P_j$.

After rejecting $n_1, \ldots, n_{i-1}$, the remaining unnormalized probability is:

$$S_{i-1} = 1 - \sum_{j=1}^{i-1} P_j.$$

The normalized probability of $n_i$ in this residual distribution becomes:

$$\hat{P}_i = \frac{P_i}{S_{i-1}}.$$

The probability of reaching $n_i$ without accepting any of the previous $i - 1$ nodes is:

$$\mathbb{P}(\text{reaching } n_i) = \prod_{j=1}^{i-1} (1 - \hat{P}_j).$$

However, since:

$$\prod_{j=1}^{i-1} (1 - \hat{P}_j) = \prod_{j=1}^{i-1} \left(1 - \frac{P_j}{S_{j-1}}\right) = \frac{S_1}{S_0} \cdot \frac{S_2}{S_1} \cdots \frac{S_{i-1}}{S_{i-2}} = \frac{S_{i-1}}{S_0} = S_{i-1},$$

and $S_0 = 1$, this implies:

$$\mathbb{P}(\text{reaching } n_i) = S_{i-1}.$$

Therefore, the total probability of accepting $n_i$ is:

$$\mathbb{P}(\mathcal{A}_i) = \mathbb{P}(\text{reaching } n_i) \cdot \hat{P}_i = S_{i-1} \cdot \frac{P_i}{S_{i-1}} = P_i.$$

**Conclusion.**    By induction, for every $i \in \{1, \ldots, k\}$, the probability of node $n_i$ being selected is exactly $P_i$. Hence, the sampling algorithm yields a sample from the original distribution $P$:

$$\mathbb{P}(\mathcal{A}_i) = P_i \quad \forall i.$$

This proves that the sequential rejection-normalization sampling procedure preserves the target distribution.

# D   GENERATION QUALITY EVALUATION: A COMPARISON BETWEEN PTD AND AUTOREGRESSIVE DECODING UNDER THE SAMPLING STRATEGY

| Benchmark | Model | Rouge-1 | Rouge-2 | Rouge-L | BLEU |
|---|---|---|---|---|---|
| MT-Bench | L-7B | 50 | 32 | 34 | 17 |
| | L-13B | 51 | 34 | 36 | 19 |
| | Q-7B | 42 | 20 | 24 | 21 |
| | Q-14B | 48 | 22 | 24 | 18 |
| | Q-32B | 48 | 24 | 26 | 22 |
| GSM-100 | L-7B | 68 | 53 | 55 | 39 |
| | L-13B | 65 | 50 | 53 | 36 |
| | Q-7B | 49 | 31 | 34 | 26 |
| | Q-14B | 52 | 29 | 31 | 28 |
| | Q-32B | 58 | 40 | 41 | 38 |
| HumanEval | CL-7B | 48 | 38 | 40 | 26 |
| | CL-13B | 48 | 40 | 43 | 21 |
| MBPP-100 | CL-7B | 82 | 77 | 80 | 77 |
| | CL-13B | 82 | 78 | 80 | 76 |

Table 2: Comparison of generated content between PTD and autoregressive decoding under the sampling strategy. All experimental settings are consistent with Table 1.

# E  ACCELERATION PERFORMANCE OF GREEDY DECODING STRATEGY

| Benchmark | Model | AR | SpeDe | | LADE | | Self-Draft | | PTD | |
|---|---|---|---|---|---|---|---|---|---|---|
| | | TP$_{(Std)}$ | TP$_{(Std)}$ | Imp. | TP$_{(Std)}$ | Imp. | TP$_{(Std)}$ | Imp. | TP$_{(Std)}$ | Imp. |
| MT-Bench | L-7B | $40_{\pm4.1}$ | $56_{\pm8.9}$ | 40% | $59_{\pm9.4}$ | 47% | $62_{\pm11.4}$ | 56% | $\mathbf{67}_{\pm10.8}$ | **68%** |
| | L-13B | $24_{\pm1.7}$ | $36_{\pm5.8}$ | 51% | $34_{\pm4.8}$ | 41% | $37_{\pm6.7}$ | 54% | $\mathbf{40}_{\pm6.4}$ | **67%** |
| | Q-7B | $36_{\pm4.4}$ | \ | \ | $59_{\pm12.2}$ | 65% | $55_{\pm13.4}$ | 52% | $\mathbf{70}_{\pm20.0}$ | **93%** |
| | Q-14B | $20_{\pm2.0}$ | \ | \ | $31_{\pm5.3}$ | 57% | $31_{\pm6.3}$ | 56% | $\mathbf{36}_{\pm6.8}$ | **81%** |
| | Q-32B | $10_{\pm0.6}$ | \ | \ | $16_{\pm2.7}$ | 57% | $16_{\pm3.3}$ | 62% | $\mathbf{19}_{\pm3.6}$ | **88%** |
| GSM-100 | L-7B | $44_{\pm1.0}$ | $64_{\pm5.5}$ | 45% | $74_{\pm5.9}$ | 66% | $75_{\pm6.7}$ | 68% | $\mathbf{85}_{\pm7.2}$ | **91%** |
| | L-13B | $26_{\pm0.4}$ | $39_{\pm3.4}$ | 49% | $41_{\pm3.3}$ | 58% | $44_{\pm4.6}$ | 67% | $\mathbf{49}_{\pm4.3}$ | **89%** |
| | Q-7B | $40_{\pm2.1}$ | \ | \ | $72_{\pm8.2}$ | 80% | $65_{\pm8.8}$ | 62% | $\mathbf{86}_{\pm16.4}$ | **116%** |
| | Q-14B | $22_{\pm0.6}$ | \ | \ | $37_{\pm3.7}$ | 67% | $37_{\pm4.6}$ | 69% | $\mathbf{44}_{\pm5.2}$ | **99%** |
| | Q-32B | $11_{\pm0.2}$ | \ | \ | $19_{\pm1.6}$ | 81% | $19_{\pm1.2}$ | 82% | $\mathbf{24}_{\pm2.2}$ | **125%** |
| HumanEval | CL-7B | $43_{\pm1.7}$ | $53_{\pm6.5}$ | 24% | $62_{\pm6.8}$ | 45% | $62_{\pm7.6}$ | 45% | $\mathbf{74}_{\pm8.5}$ | **74%** |
| | CL-13B | $25_{\pm0.7}$ | $34_{\pm5.3}$ | 34% | $37_{\pm4.5}$ | 45% | $39_{\pm5.3}$ | 55% | $\mathbf{44}_{\pm5.8}$ | **74%** |
| MBPP-100 | CL-7B | $45_{\pm0.8}$ | $62_{\pm5.9}$ | 39% | $77_{\pm6.4}$ | 71% | $73_{\pm7.2}$ | 62% | $\mathbf{93}_{\pm9.9}$ | **108%** |
| | CL-13B | $26_{\pm0.3}$ | $39_{\pm3.9}$ | 48% | $43_{\pm4.1}$ | 64% | $48_{\pm4.5}$ | 82% | $\mathbf{55}_{\pm5.4}$ | **107%** |

Table 3: Throughput and Improvement (Imp.) under greedy decoding for PTD, Auto-Regressive decoding (AR), the vanilla Speculative Decoding (SpeDe) method with draft model of LLaMA-68M (Miao et al., 2024), the Lookahead decoding (LADE) (Fu et al., 2024), and Self-Draft Gao et al. (2025).

| Benchmark | Model | LADE | | | | Self-Draft | | | | PTD | | | |
|---|---|---|---|---|---|---|---|---|---|---|---|---|---|
| | | DE | HR | AL | Dft/Ver | DE | HR | AL | Dft/Ver | DE | HR | AL | Dft/Ver |
| MT-Bench | L-7B | 1.95 | 0.69 | 2.39 | 59/23 | 1.96 | 0.95 | 2.02 | 30/30 | **2.23** | 0.71 | **2.74** | 35/23 |
| | L-13B | 1.83 | 0.67 | 2.26 | 39/17 | 1.96 | 0.95 | 2.02 | 30/30 | **2.20** | 0.71 | **2.70** | 34/22 |
| | Q-7B | 2.20 | 0.78 | 2.55 | 59/31 | 2.03 | 0.92 | 2.12 | 31/26 | **2.58** | 0.80 | **2.99** | 40/29 |
| | Q-14B | 2.01 | 0.76 | 2.31 | 39/21 | 1.97 | 0.92 | 2.05 | 31/26 | **2.40** | 0.80 | **2.76** | 42/29 |
| | Q-32B | 1.87 | 0.72 | 2.21 | 27/15 | 2.02 | 0.92 | 2.11 | 31/25 | **2.43** | 0.77 | **2.86** | 37/27 |
| GSM-100 | L-7B | 2.23 | 0.72 | 2.72 | 58/22 | 2.29 | 0.94 | 2.38 | 30/32 | **2.52** | 0.73 | **3.09** | 32/21 |
| | L-13B | 2.06 | 0.70 | 2.53 | 38/16 | 2.29 | 0.94 | 2.38 | 30/32 | **2.48** | 0.72 | **3.05** | 31/20 |
| | Q-7B | 2.44 | 0.83 | 2.75 | 59/35 | 2.25 | 0.95 | 2.32 | 31/28 | **2.90** | 0.84 | **3.26** | 37/31 |
| | Q-14B | 2.16 | 0.80 | 2.45 | 39/23 | 2.19 | 0.95 | 2.25 | 31/28 | **2.68** | 0.84 | **3.00** | 41/32 |
| | Q-32B | 2.16 | 0.80 | 2.46 | 27/17 | 2.34 | 0.96 | 2.40 | 31/29 | **2.91** | 0.84 | **3.28** | 35/30 |
| HumanEval | CL-7B | 1.96 | 0.67 | 2.44 | 58/20 | 2.15 | 0.94 | 2.24 | 30/30 | **2.35** | 0.69 | **2.97** | 33/19 |
| | CL-13B | 1.95 | 0.66 | 2.45 | 38/15 | 2.23 | 0.93 | 2.33 | 30/29 | **2.35** | 0.68 | **3.00** | 31/17 |
| MBPP-100 | CL-7B | 2.29 | 0.71 | 2.82 | 58/23 | 2.47 | 0.94 | 2.57 | 30/30 | **2.75** | 0.74 | **3.36** | 33/20 |
| | CL-13B | 2.13 | 0.69 | 2.63 | 38/16 | 2.48 | 0.94 | 2.59 | 30/30 | **2.72** | 0.74 | **3.34** | 32/19 |

Table 4: Decoding Efficiency (DE), hit rate (HR), Accept Length (AL) and overheads (Dft/Ver) of PTD, LADE (Fu et al., 2024), and Self-Draft (Gao et al., 2025) under greedy decoding strategy.

