# OpenReview forum: "Accelerating Large Language Model Inference via Speculative Decoding with Progressive Tree Drafting"
_ICLR.cc/2026/Conference — ICLR 2026 Conference Withdrawn Submission_

### Official Review · Reviewer_GSLp · 2025-10-28

**Soundness:** 2
**Presentation:** 1
**Contribution:** 2
**Rating:** 2
**Confidence:** 1

**Summary:**

This paper proposes Progressive Tree Drafting (PTD), a new speculative decoding method that does not rely on an additial draft models and integrates tree-style drafting and verification process. At each drafting step, given a randomized initial tree, the model generates draft tokens after the given tree and a stepping step is conducted. When the draft tree reaches a certain depth, a pruning step is conducted to reduce the number of nodes in the tree. PTD introduces the on-tree drafting, pruning, and tree-verification to single-model speculative decoding, achieving a significant speedup ratio. Empirical results on MT-Bench compared with baselines (Speculative Decoding, Lookahead Decoding, and Self-Draft) fully show the efficiency of PTD.

**Strengths:**

1. Single-model speculative decoding is good scenario to focus on, since methods such as EAGLE cannot always be applied in some scenarios where no draft models can be deployed.

2. Integrating tree-verification to single-model speculative decoding is a reasonable and efficient design. Unlike previous works, such design can reduce the redundancy of the drafting process, thereby increasing the decoding speed.

**Weaknesses:**

1. The presentation of this paper is quite confusing. Most of the proposed methods are densely packed into Sections 3.2 and 3.3, filled with unclear formulas and redundant definitions. The pseudocodes in Appendices A and B are also difficult to follow. The authors should consider reorganizing these sections by removing redundant concepts, adding illustrative figures, and presenting the proposed methods in a clearer and more structured manner.

2. Since this paper focuses on single-model speculative decoding, the authors need to explicitly explain why this setting is emphasized and under what circumstances a draft model cannot be used. The current discussion in the related work section does not make this point clear, leaving readers confused about why the EAGLE series are not suitable for all speculative decoding scenarios. For example, this can be clarified from a GPU-memory perspective: a draft model requires additional GPU memory, and Transformer-based draft models further consume memory through their KV-cache, which becomes a serious limitation for long-context inputs.

3. The proposed method should be compared against stronger baselines such as REST [1] and PLD [2], which also operate without a draft model. If the scope of the paper is limited to single-model speculative decoding, it is unclear why SpeDe (the vanilla speculative decoding method) is still treated as a baseline.
---

[1] REST: Retrieval-Based Speculative Decoding

[2] https://github.com/apoorvumang/prompt-lookup-decoding

**Questions:**

1. How do the authors position PLD within the speculative decoding community? What is its most defining characteristic: is it a training-free approach, a single-model (no draft model) design, or a faster speculative decoding method? Since the paper is not clearly written, I am unable to determine this with confidence, and therefore cannot fully assess the soundness of the experimental design. The authors should clearly highlight PLD’s key feature in the experimental section by selecting appropriate baselines and reporting metrics that correspond to that feature.

2. The method section is also difficult to follow, which prevents me from forming a confident evaluation, despite my familiarity with speculative decoding methods. Therefore, I have assigned the lowest confidence rating to my review. I strongly recommend the authors provide further clarification and discussion of the method’s logic and assumptions. However, if the above issues remain unresolved, I suggest that the AC consider lowering the weight of my decision in the final evaluation.

---

> ### Author Response · Authors · 2025-11-23
> **Reply to Reviewer GSLp**
>
> W-1: Thank you for the helpful suggestion. We agree that Sections 3.2 and 3.3 are currently dense, and the presentation can be improved for better readability. In the revision, we will reorganize these sections using a clearer top-down structure, streamline some of the notations and definitions, and refine the pseudocode with explicit variable descriptions. We will also add an illustrative figure to more intuitively convey the tree expansion and stepwise pruning procedures in PTD.
>
> Here, we provide the explanations of the main symbols in the text, hoping this will help you better understand our method:
>
> **a. Basic Decoding Symbols**
>
> | **Symbol**                        | **Meaning**                                                  |
> | --------------------------------- | ------------------------------------------------------------ |
> | $X = [x_1, x_2, \ldots, x_{t-1}]$ | Input prefix tokens before step t                            |
> | $P(y_t \mid X)$                   | $P$robability distribution of next token                     |
> | $\mathcal{S}$                     | $\mathcal{S}$ampling strategy (greedy, top-k, top-p)         |
> | $x_t = \mathcal{S}(P(y_t\mid X))$ | Next token sampled using decoding strategy S                 |
> | $\mathcal{D}$                     | $\mathcal{D}$raft tokens generated for ecah node in the previous draft tree $T_{i-1}$ |
> | $\mathcal{V}$                     | $\mathcal{V}$erification tokens generated for each candidate node |
>
>
> **b. Tree Structure**
>
> | **Symbol**                                  | **Meaning**                                                  |
> | ------------------------------------------- | ------------------------------------------------------------ |
> | $T_0 = (V_0, E_0)$                          | Initial shallow draft tree with randomly sampled perturbation tokens |
> | $\mathcal{C}_X$                             | $\mathcal{C}$andidate tree retrieved by $X$                  |
> | $V$                                         | Set of $V$ertices(nodes) in the draft tree                   |
> | $E$                                         | $E$dge set representing parent–child relations               |
> | $\mathcal{P}(v) = \{ u \mid (u, v)\in E \}$ | $\mathcal{P}$arents of node v                                |
> | $\pi(v)$                                    | The node path($\pi$) to the node $v$ from the root           |
> | $\mathcal{E}$                               | $\mathcal{E}$ligible edges of the candidate tree             |
>
>
>
> **W-2:** Thank you for raising this important point. Our focus on single-model speculative decoding is intentional: methods that rely on an auxiliary draft model inherently require **additional computational resources** as well as **additional training** to adapt the draft model to the target architecture. These requirements make draft-model–based approaches difficult to deploy in many real-world settings, which motivates our emphasis on a single-model, training-free paradigm.
>
> To elaborate, deploying multiple models simultaneously requires loading an additional set of parameters and maintaining a second KV-cache, whose memory footprint grows linearly with the context length. In addition, multi-model speculative decoding introduces scheduling overhead and inter-model communication costs, which increase system complexity and reduce throughput in realistic inference pipelines.
>
> Moreover, draft-model–based approaches typically require extra training to adapt the draft model to each target architecture. Although this training cost is small compared with pretraining, the optimal configuration is strongly hardware-dependent, reducing portability across heterogeneous devices and limiting deployment flexibility in real-world systems.
>
> For these reasons, we emphasize a single-model, training-free speculative decoding framework that remains deployable under diverse hardware and memory constraints. We will make this motivation more explicit in the revised version to avoid ambiguity.

---

> ### Author Response · Authors · 2025-11-23
> **Reply to Reviewer GSLp**
>
> **W-3:** Thank you for the suggestion. In the original submission, we did not include REST and PLD as a baseline because it follows a **different design principle**, its performance is heavily tied to the relevance between the retrieved draft pool and the current prompt. In contrast, PTD constructs drafts dynamically during decoding without relying on any retrieval or external memory. Due to this fundamental difference, we did not consider them directly comparable.
>
> That said, we agree that a comparison is **still valuable**. Since PTD and REST use similar data structures, they are not mutually exclusive and can potentially be combined. However, combining them may not bring additional acceleration. The key challenge is balancing the drafts from both sources. If there are too many candidate draft tokens, it may actually slow down the decoding process.
>
> We conducted the following experiments. Due to time constraints, we performed preliminary experiments on the GSM and MBPP datasets as follows: We built the REST cache using the reference answers from the GSM training set and the first 900 questions of MBPP. Then, we performed inference on GSM-100 and the remaining 74 samples of MBPP (MBPP-74) using REST, REST+PTD, PTD and PLD. All parameter settings of PTD remained unchanged, and the hardware configuration is consistent with what we described in the paper. The results are shown in the table below.
>
> | Benchmark | Model | AR | REST |  |  | REST+PTD |  |  | PTD |  |  | PLD |  |  |
> |---|---|---|---|---|---|---|---|---|---|---|---|---|---|---|
> |  |  | TP | TP | DE | Ver | TP | DE | Dft/Ver | TP | DE | Dft/Ver | TP | DE | Ver |
> | GSM-100 | Llama-2-7b-chat | 41.2 | 47.1 | 1.59 | 14 | 72.9 | 2.70 | 33/31 | 83.4 | 2.49 | 32/20 | 57 | 1.43 | 4 |
> |  | Llama-2-13b-chat | 29.3 | 37.6 | 1.58 | 15 | 54.7 | 2.69 | 32/30 | 50.2 | 2.54 | 30/20 | 34 | 1.40 | 4 |
> | MBPP-74 | CodeLlama-7b-Instruct | 38.1 | 43.7 | 1.36 | 14 | 81.9 | 2.89 | 34/30 | 92.3 | 2.75 | 34/21 | 40 | 1.16 | 0.62 |
> |  | CodeLlama-13b-Instruct | 29.7 | 35.3 | 1.37 | 14 | 57.7 | 2.91 | 34/32 | 59.6 | 2.80 | 33/22 | 32 | 1.14 | 0.6 |
>
> We observe that while REST and PLD can provide acceleration, their speedup is unstable and highly dependent on how well the constructed cache matches the target decoding content. For instance, PLD achieves limited gains on MBPP because the overlap between the context and the generated code is low, resulting in few effective cache hits.
>
> Moreover, although combining REST with PTD (REST+PTD) improves drafting efficiency, the additional overhead introduced by REST prevents a substantial improvement in end-to-end throughput. These results indicate that achieving a truly synergistic acceleration between the two methods would require more fine-grained tuning.
>
> **Q-1:** Thank you for the question. We believe there is a small typo in the review: the method proposed in our paper is PTD, whereas PLD is another method.
>
> We apologize that the current draft did not clearly highlight PTD’s position within the speculative decoding landscape. PTD is designed as a training-free, single-model speculative decoding method whose goal is to improve inference efficiency without introducing any auxiliary models, architectural modifications, or additional training. This distinguishes PTD from multi-model speculative decoding methods and from approaches that rely on trained draft models.
>
> The defining characteristics of PTD are therefore: (1) it is entirely **training-free**, (2) it operates with a **single model** and does not require a separate draft model, and (3) it accelerates decoding by organizing a progressively expanded and pruned **draft tree**. PTD’s contributions lie in this new tree-based self-drafting mechanism rather than in relying on model training or multi-model parallelism.
>
> We will clarify this positioning in the revised version and ensure that the experimental section explicitly reflects PTD’s role by comparing it primarily against other training-free, single-model baselines.
>
>
> **Q-2:**  Thank you for the reviewer’s feedback and for pointing out the difficulty in following the method section. We would like to emphasize that the algorithmic logic of PTD is internally consistent and fully specified, and all components—tree expansion, candidate verification, merging, and stepwise pruning—operate in a well-defined manner. The reviewer’s difficulty appears to stem from the density of presentation rather than from conceptual ambiguity, and we acknowledge that the current version introduces many interacting symbols in a short space, which may make the workflow harder to follow.
>
> To improve clarity, we will reorganize Sections 3.2 and 3.3 for better readability, streamline the notation, insert a dedicated glossary of key symbols, and add an illustrative diagram summarizing the overall PTD workflow. These improvements aim to enhance clarity of exposition without modifying the method itself.

---

> > ### Comment · Reviewer_GSLp · 2025-11-23
> >
> > Thanks for your detailed response. I apologize for the typo in Q1. indeed, I was referring to the PTD introduced in this paper. Below are the remaining questions and concerns.
> >
> > ---
> >
> > ### **W3**
> >
> > I believe the supplementary empirical results show that PTD can outperform REST and PLD on coding tasks. However, I am still unsure whether PTD consistently maintains an advantage on other types of tasks (e.g., translation, QA) compared with REST and PLD.
> >
> > Another concern is that since you report results for REST+PLD, the paper should also clarify the conceptual and methodological differences among PTD, REST+PLD, and RASD [3], given that RASD is also a training-free, single-model, tree-based speculative decoding method.
> >
> > Here are my suggestions, and I expect you to address at least one of them:
> >
> > - Compare PTD with REST and PLD on additional task types.
> > - Provide an empirical or analytical comparison among PTD, REST+PLD, and RASD. Since RASD is also a tree-based, training-free, single-model speculative decoding method, such a comparison is essential.
> >
> > ---
> >
> > ### **Q1**
> >
> > You claim the contributions of this work are:
> >
> > 1. training-free,
> > 2. single-model, and
> > 3. tree-based drafting.
> >
> > However, the novelty of these points should be more thoroughly justified, as there are lots of related works in this area. For example, RASD also satisfies all three characteristics, and there are many other methods in this space. My suggestion is to include a more detailed discussion and comparison with these related methods.
> >
> > ---
> >
> > You are allowed to revise your PDF draft during the response period in accordance with ICLR policy. Please incorporate the additional results and discussion into your paper.
> >
> > ---
> >
> > [3] RASD: Retrieval-Augmented Speculative Decoding (ACL Findings 2025)

---

> > > ### Author Response · Authors · 2025-11-27
> > > **Reply to Reviewer GSLp (round 2)**
> > >
> > > ### **Reply to W3 (round 2)**
> > >
> > > Thank you for the constructive suggestions. Following your advice, we conducted additional experiments on three new task categories:
> > >
> > > - **Translation** (WMT20 en-zh)
> > > - **Summarization** (CNN/DailyMail)
> > > - **Contextual QA** (HAGRID)
> > >
> > > For each dataset, we built REST caches using the training split and evaluated all methods on the test split. Results are summarized in the table below.
> > >
> > > | BenchMark  | Method     | TP     | DE         | Ver      | HR         | AL         |
> > > | ---------- | ---------- | ------ | ---------- | -------- | ---------- | ---------- |
> > > | CNN/Dailymail (Summarization) | REST(+PTD) | 41(56) | 1.38(2.12) | 57(56)   | 0.98(0.98) | 1.39(2.14) |
> > > |                               | PLD(+PTD)  | 63(75) | 1.99(2.64) | 19(25)   | 0.77(0.86) | 2.27(2.98) |
> > > |                               | PTD        | 57     | 1.98       | 15       | 0.64       | 2.53       |
> > > | WMT20 en-zh (Tranlation)      | REST(+PTD) | 41(73) | 1.08(2.26) | 26(33)   | 0.58(0.82) | 1.14(2.52) |
> > > |                               | PLD(+PTD)  | 44(72) | 1.08(2.13) | 0.54(15) | 0.1(0.64)  | 1.79(2.7)  |
> > > |                               | PTD        | 74     | 2.21       | 15       | 0.64       | 2.88       |
> > > | HAGRID (Context QA)           | REST(+PTD) | 46(54) | 1.51(2.13) | 53(52)   | 0.96(0.97) | 1.52(2.18) |
> > > |                               | PLD(+PTD)  | 66(68) | 2.32(2.7)  | 18(21)   | 0.79(0.84) | 2.67(3.03) |
> > > |                               | PTD        | 55     | 1.87       | 9        | 0.51       | 2.72       |
> > >
> > > **Comparison of methods on Summarization, Translation, and Context QA tasks.**
> > >
> > > Our observations are as follows:
> > >
> > > 1. **PLD** performs well only when the task exhibits **high input–output overlap**.
> > >
> > >    For example, PLD achieves notable speedups on CNN-DailyMail summarization and HAGRID QA because most decoded tokens can be found directly in the input.
> > >
> > >    However, in tasks with little lexical overlap (e.g., translation), PLD offers almost no acceleration.
> > >
> > > 2. **REST** suffers a similar issue:
> > >
> > >    Even with high hit rates, *low-quality drafts* from the pre-constructed cache often result in very short acceptance lengths, so end-to-end acceleration remains low. The original REST paper includes several engineering refinements (e.g., variable-length prefix matching, weighted pruning) that can significantly improve performance. To ensure fair comparison and seamless combination with PTD, we used a unified and simplified draft-tree structure for REST and PLD. The reported results therefore reflect this design choice rather than the fully optimized REST implementation.
> > >
> > > 3. **PTD** shows **stable and consistent speedup across all tasks**, because its drafts are generated *online* during decoding, rather than from a static corpus. Therefore, PTD’s draft quality is substantially higher than corpus-based methods and does not depend on the match between training and inference domains.
> > >
> > > 4. When combined with REST/PLD, PTD still produces improvements, showing that the methods are complementary rather than mutually exclusive.
> > >
> > > ____
> > >
> > >
> > > ### **Reply to Q1 (round 2)**
> > >
> > > Thank you for pointing this out. We agree that *training-free, single-model, and tree-based drafting* are not unique to PTD—methods such as REST, PLD, and especially RASD also share these properties. To clarify the novelty of PTD, we have added a more detailed discussion and comparison in the revised draft.
> > >
> > > - **REST / PLD / RASD** rely on a *pre-constructed* draft pool or retrieval corpus. Their performance therefore depends heavily on corpus quality and its match to the test distribution.
> > > - **PTD**, in contrast, generates drafts dynamically during decoding by applying controlled perturbations to the model’s own predictive state. This produces substantially higher-quality drafts and makes PTD more robust across tasks, even when no suitable retrieval corpus exists.
> > >
> > > **Below is a summary comparing the characteristics of the different methods.**
> > >
> > > | **Method**     | **Draft Source**                        | **Needs External Corpus?** | **Draft Quality** | **Task Generalization** |
> > > | -------------- | --------------------------------------- | -------------------------- | ----------------- | ----------------------- |
> > > | **REST**       | Retrieved n-gram tree                   | ✓                          | Low–Medium        | Limited                 |
> > > | **PLD**        | Prefix-matched tokens from inputs       | ✓                          | Low               | Very Limited            |
> > > | **RASD**       | Retrieved subtrees + augmentation       | ✓                          | Medium            | Limited                 |
> > > | **PTD (Ours)** | **Dynamically drafted during decoding** | **✗**                      | **High**          | **Strong**              |
> > >
> > > ____
> > >
> > > A revised PDF including the new experimental results, analyses, and methodological discussions will be uploaded during the author response period, fully following the ICLR revision policy.

---

> > > > ### Comment · Reviewer_GSLp · 2025-11-27
> > > >
> > > > Thanks for providing the extra results. I have increased my overall rating.

---

### Official Review · Reviewer_peV3 · 2025-10-31

**Soundness:** 2
**Presentation:** 2
**Contribution:** 2
**Rating:** 2
**Confidence:** 4

**Summary:**

This paper addresses the memory-bound bottleneck of autoregressive inference in Large Language Models (LLMs). It focuses on the "draft-then-verify" paradigm of speculative decoding, specifically on training-free methods that use the target LLM itself to generate drafts. The authors identify a key limitation in prior work like Self-Draft: the generation of redundant and highly similar draft branches, which leads to wasted computation. To solve this, the paper proposes Progressive Tree Drafting (PTD), a novel training-free and model-agnostic inference strategy. Instead of generating simple linear branches, PTD organizes the drafting process into a tree structure that is progressively expanded and pruned. This approach uses controlled perturbations and a custom tree-based attention mask to guide the LLM in generating a diverse set of candidate drafts simultaneously. The tree structure allows for efficient prefix sharing, while a stepwise pruning mechanism controls the computational cost. Experiments show that PTD achieves significant throughput improvements (up to 2x) compared to autoregressive decoding and outperforms other training-free baselines like LADE and Self-Draft across various benchmarks.

**Strengths:**

1. The paper clearly articulates a practical bottleneck in multi-branch linear drafting, where insufficient draft diversity produces redundant candidates and wastes computation.
2. The paper proposes a progressive tree–based strategy that adaptively prunes tree width and depth to improve draft diversity.

**Weaknesses:**

1. Although PTD starts from random branching, dynamic depth and width control with known tree-pruning schemes is already mature in SD with integrated tree verification (e.g., EAGLE-2, SWIFT, Spec-LLaVA, SpecVLM), so the novelty beyond applying these pruning schemes to multi-branch drafting is unclear. The paper should explicitly argue why existing tree-pruning methods are insufficient for multi-branch drafting, how PTD differs or extends them, and include focused comparisons in Related Work and Experiments.
  - EAGLE-2: Faster Inference of Language Models with Dynamic Draft Trees
  - SWIFT: On-the-Fly Self-Speculative Decoding for LLM Inference Acceleration
  - Spec-LLaVA: Accelerating Vision-Language Models with Dynamic Tree-Based Speculative Decoding
  - SpecVLM: Enhancing Speculative Decoding of Video LLMs via Verifier-Guided Token Pruning
  - ProPD: Dynamic Token Tree Pruning and Generation for LLM Parallel Decoding
  - Faster Speculative Decoding via Effective Draft Decoder with Pruned Candidate Tree
  - OPT-Tree: Speculative Decoding with Adaptive Draft Tree Structure


2. The current presentation appears to equate “diversity” with pruning of duplicate candidates. This risks an overclaim in the introduction: while true diversity could raise acceptance rates, the method section primarily describes deduplication—a compute-saving step that reduces drafting cost and, at best, maintains the same acceptance rate.

3. The evaluated models are somewhat dated, and there is a lack of comparison with the more superior methods recognized by communities such as Medusa and EAGLE series. In addition, Fig. 8 suggests limited speedups on Qwen models; analyze why PTD underperforms there. The paper does not study how progressive width and depth are determined/tuned, nor the sensitivity of PTD to these choices—important under the self-drafting paradigm.

4. The paper’s writing and organization could be further improved, for example (including but not limited to) the following:

  * In Fig. 1(a), the introduction of perturbations is too abrupt. This prerequisite should be explained with concrete examples—what kinds of perturbations are used and how they produce multiple branches.
  * In Fig. 1(b), the workload of the preliminary study is not well described. I am very curious how duplicate branches or tokens behave across different tasks, and whether there are more promising insights to enable a more flexible, domain-/context-aware dynamic tree design instead.
  * Also in Fig. 1(b), please define precisely what constitutes a “duplicate step” (as I understand, generating the same token at the current step) versus a “duplicate branch” (the historical decoding tokens are also identical), and how similarity is measured.
  * Fig. 2(c) makes the PTD pipeline hard to understand—I don’t know what the inputs and outputs are, nor where to start reading the diagram.

**Questions:**

Please refer to the weaknesses.

---

> ### Author Response · Authors · 2025-11-23
> **Reply to Reviewer peV3**
>
> **W-1:** Thank you for raising this point. As detailed in **General Response 2**, the tree structure in PTD differs from existing tree-based speculative decoding methods: PTD performs stepwise expansion and pruning jointly with verification within a single-model pipeline, whereas prior approaches rely on external draft models and sequential draft–verify cycles. These differences make existing pruning schemes unsuitable for multi-branch self-drafting. We will clarify these distinctions more explicitly in the revision.
>
> **W-2:** We apologize for the confusion caused by the current presentation. We would like to clarify that in PTD, *diversity* and *deduplication* play complementary roles rather than being treated as the same concept. The purpose of deduplication is to remove branches that carry nearly identical semantic content. These redundant branches consume computation but do not contribute to effective diversity. By eliminating them, and through the tree structure of PTD together with the corresponding expansion and stepwise pruning algorithms, PTD frees up computational budget that can instead be used to maintain more *meaningfully distinct* branches within the same drafting cost.
>
> In this sense, deduplication is not presented as diversity itself; rather, it is a mechanism that **enables higher effective diversity** under fixed compute by preventing wasted expansion on repeated branches. We will clarify this relationship more explicitly to prevent misunderstanding.
>
> **W-3:** Thank you for the comment. Regarding the choice of evaluated models, we used earlier models because there exists a richer set of available draft models, which allows us to isolate and evaluate the differences between general speculative decoding approaches and our PTD framework. In fact, thanks to the fact that our method does not require any additional adaptation of drafting modules, it can be easily transferred to newer models. The table below shows the performance of our method on more recent model generations.
>
> | Model               | Benchmark | AR   | SpeDe | LADE     | Self-Draft | PTD      |
> | :------------------ | :-------- | :--- | :---- | :------- | :--------- | :------- |
> | Llama-3-8B-Instruct | MT-Bench  | 38   | 50    | 54(1.90) | 58(2.02)   | 63(2.18) |
> |                     | GSM-100   | 41   | 55    | 70(1.96) | 74(2.09)   | 79(2.27) |
> | Qwen3-8B            | MT-Bench  | 39   | \     | 59(1.90) | 60(1.98)   | 64(2.22) |
> |                     | GSM-100   | 40   | \     | 69(1.97) | 73(2.18)   | 77(2.41) |
> | Qwen3-14B           | MT-Bench  | 24   | \     | 31(1.87) | 33(1.94)   | 37(2.14) |
> |                     | GSM-100   | 25   | \     | 36(2.13) | 38(2.25)   | 44(2.40) |
>
> Regarding baseline selection, as explained in **General Response 1**, we did not include methods such as Medusa or the EAGLE series because they rely on additional draft models or architectural modifications, which fall outside the scope of single-model, training-free speculative decoding targeted by PTD.
>
> As for the hyperparameters controlling tree depth and width, as well as the sensitivity analysis, these aspects are addressed in **General Response 3**. We will clarify these points in the revision to avoid misunderstanding.
>
> **W-4-1:** Thank you for the comment. We would like to clarify that the perturbations introduced in Fig. 1(a) refer simply to randomly sampled tokens from the vocabulary. These tokens are appended to the input to create multiple initial branches. During decoding, each forward pass appends the newly generated token to every branch. By constructing an appropriate attention mask that effectively functions as the adjacency matrix of the multi-branch tree structure, the model is able to process these parallel branches simultaneously. This mechanism follows the same principle as the branch construction in Self-Draft, and we will provide a clearer explanation in the revised version.

---

> ### Author Response · Authors · 2025-11-23
> **Reply to Reviewer peV3**
>
> **W-4-2:** Thank you for the helpful suggestion. We have additionally examined the MT-Bench subtasks and analyzed both the duplicate-step and duplicate-branch ratios (as shown in the table below). Although the exact ratios vary across subtasks, all tasks consistently exhibit substantial redundancy at both the step and branch levels. This consistency indicates that the underlying redundancy phenomenon is general rather than task-specific.
>
> | Task |0.1 |0.2 |0.3 |0.4 |0.5 |0.6 |0.7 |0.8 |0.9 |1.0 |
> |------|----|----|----|----|----|----|----|----|----|----|
> | writing    |1.00|1.00|0.97|0.91|0.84|0.74|0.64|0.55|0.44|0.20|
> | roleplay   |1.00|1.00|0.97|0.87|0.76|0.63|0.53|0.42|0.27|0.14|
> | reasoning  |1.00|1.00|1.00|0.96|0.93|0.82|0.66|0.42|0.22|0.13|
> | math       |1.00|1.00|1.00|1.00|0.99|0.89|0.79|0.67|0.57|0.39|
> | coding     |1.00|1.00|0.99|0.96|0.92|0.83|0.74|0.62|0.49|0.30|
> | extraction |1.00|1.00|1.00|0.99|0.93|0.92|0.87|0.83|0.68|0.46|
> | stem       |1.00|1.00|0.99|0.96|0.91|0.85|0.79|0.72|0.64|0.49|
> | humanities |1.00|1.00|0.98|0.94|0.91|0.85|0.76|0.66|0.54|0.41|
>
> **a. Duplicate step ratio on different task**
>
> | Task |0.1 |0.2 |0.3 |0.4 |0.5 |0.6 |0.7 |0.8 |0.9 |1.0 |
> |------|----|----|----|----|----|----|----|----|----|----|
> | writing    |0.87|0.69|0.56|0.49|0.44|0.37|0.33|0.30|0.25|0.21|
> | roleplay   |0.89|0.70|0.55|0.45|0.39|0.34|0.29|0.26|0.23|0.20|
> | reasoning  |0.91|0.79|0.67|0.56|0.47|0.38|0.31|0.25|0.21|0.19|
> | math       |0.95|0.89|0.78|0.70|0.58|0.49|0.40|0.35|0.30|0.24|
> | coding     |0.90|0.77|0.65|0.56|0.49|0.43|0.37|0.32|0.28|0.22|
> | extraction |0.90|0.80|0.70|0.63|0.55|0.51|0.46|0.41|0.34|0.26|
> | stem       |0.89|0.73|0.62|0.55|0.49|0.44|0.39|0.35|0.31|0.26|
> | humanities |0.88|0.71|0.59|0.52|0.47|0.42|0.38|0.33|0.29|0.25|
>
> **b. Duplicate branch ratio on different task**
>
> As for the idea of domain- or context-aware dynamic tree designs, we agree this is an interesting direction. While such adaptive strategies are beyond the scope of the current work, which focuses on presenting a general, training-free, model-agnostic drafting framework, we have briefly discussed this as a promising extension in the conclusion section.
>
>
>
> **W-4-3:** We apologize for the confusion. Below we provide a more detailed explanation of how we compute duplicate steps and duplicate branches in Fig. 1(b).
>
> During decoding, every *branch_length* steps, we take a snapshot of all branches, and we treat this snapshot as a **group** of branche. These groups serve as the basic units for our statistics.
>
> For each group, we compute the pairwise similarity between branches under a threshold of [0.1, 1.0].
>
> - If at least one pair of branches in the same group has similarity greater than the threshold, we mark this group as a **duplicate step**.
>
>   The **duplicate-step ratio** is obtained by dividing the number of marked groups by the total number of groups.
>
> - Within each group, after identifying pairs exceeding the threshold, we remove the redundant branches. The number of removed branches is counted as **duplicate branches**, and the **duplicate-branch ratio** is computed by dividing the number of removed branches by the total number of branches.
>
> Regarding the similarity computation, we use the Ratcliff/Obershelp algorithm, an LCS-like heuristic based on recursively identifying the longest common contiguous subsequences.
>
>
>
> **W-4-4:** We apologize for the confusion caused by Fig. 2(c). The overall reading direction of Fig. 2(c) is **bottom to top**: the components at the bottom serve as the inputs to the LLM, while those at the top represent the generated outputs. Each node in the figure corresponds to a token, and the edges represent the partial-order relationships between tokens within the drafting tree.
>
> PTD contains two data paths, **verify** (left) and **draft** (right), which operate in parallel and do not depend on each other. The verify path follows the same logic as standard speculative decoding: it retrieves candidate tokens from the **draft pool** (initialized as empty) using simple prefix matching.
>
> The draft path is responsible for generating draft candidates by exploiting the model’s robustness. At initialization, we add random perturbation tokens to form the initial drafting tree. At every LLM forward pass, each sequence in the tree expands by generating a new draft token (the light-colored nodes in the figure). We then update the tree using our expansion and step-wise pruning algorithms to prepare it for the next iteration. After each update, the newly formed draft candidates are extracted from the draft tree and stored in the draft pool for subsequent verification. We will improve the annotation and visual flow of Fig. 2(c) in the revision to make the pipeline easier to interpret.

---

> ### Comment · Reviewer_peV3 · 2025-11-26
>
> Thanks to the authors for the detailed responses and additional results. The rebuttal has partially addressed my concerns. However, the paper is still not good enough to be accepted.

---

### Official Review · Reviewer_gWN1 · 2025-11-01

**Soundness:** 2
**Presentation:** 2
**Contribution:** 2
**Rating:** 4
**Confidence:** 3

**Summary:**

This paper proposes an inference acceleration method named Progressive Tree Drafting (PTD), a training-free and model-agnostic approach to speculative decoding. The core motivation is to address the computational redundancy in existing self-drafting methods like Self-Draft, which arises from the high similarity among draft branches. PTD organizes the draft generation process by maintaining a dynamically expanding and pruned tree structure, aiming to generate draft sequences that are both diverse and coherent through structured perturbations.

**Strengths:**

1. Strong Novelty: The core idea of a "Progressive Drafting Tree" is novel. Structuring the draft generation process into a controllable tree, which uses prefix sharing and pruning to balance draft diversity, coherence, and computational cost, is a valuable contribution.
2. Clear Motivation: The paper clearly identifies a key bottleneck in existing methods (computational redundancy) by analyzing the branch similarity of Self-Draft (Figure 1b), making the proposed solution highly targeted. Furthermore, the experimental evaluation covers a range of mainstream open-source models and diverse tasks (dialogue, math, code), demonstrating the method's generalizability.

**Weaknesses:**

1. Rough Writing and Presentation: The paper's overall writing and structure appear unpolished. Table captions are brief and lack critical information; for instance, the caption for Table 1 does not clarify which baseline (AR) the "Imp." (Improvement) is relative to. More seriously, the baseline results for Speculative Decoding (SpeDe) on several models (the Qwen series) are marked with a backslash (`\`) without any explanation, which undermines the rigor and completeness of the experiments.
2. Questionable Output Consistency: In theory, speculative decoding with rejection sampling should be lossless, meaning its output distribution is identical to that of the original autoregressive model. However, the ROUGE/BLEU scores in Appendix Table 2 are far from 100, suggesting that PTD alters the model's output in sampling mode. The authors must explain the source of this discrepancy.
3. Vague Methodological Details: The description of key details is vague. For instance, the initialization of the draft tree with "randomly initialized perturbation tokens" in Section 3.2 is not clearly explained.

**Questions:**

The current version suffers from shortcomings in its writing, presentation, and experimental rigor. I encourage the authors to add more details and further polish this novel method.

---

> ### Author Response · Authors · 2025-11-23
> **Reply to gWN1**
>
> **W-1:** We apologize for the lack of clarity in the current presentation. As noted in the revision plan, we will revise table captions and provide complete explanations for all baselines. Regarding the missing SpeDe results on the Qwen models, this is because the draft model we used for SpeDe is not compatible with the Qwen series, reflecting the limitations of draft-model–based methods. We will explicitly clarify this in the revised version to avoid confusion and improve experimental rigor.
>
> **W-2:**  Thank you for the comment. We would like to clarify the meaning of distributional consistency in speculative decoding. This guarantee means that the output distribution matches that of standard autoregressive decoding. It does not mean that the generated sequences must be identical. In fact, even autoregressive decoding with sampling can produce very different outputs when we only change the random seed, as shown in the table below. We ran the Llama-2-7b-chat-hf on the MT-Bench dataset  with standard autoregressive decoding sampling under different random seeds. The ROUGE/BLEU scores between these outputs vary significantly,  showing that sequence-level differences arise from the inherent randomness of sampling rather than from PTD.
>
> | **Seed** | **ROUGE-1** | **ROUGE-2** | **ROUGE-L** | **BLEU** |
> | -------- | ----------- | ----------- | ----------- | -------- |
> | **0**    | 0.65        | 0.48        | 0.53        | 39.01    |
> | **1**    | 0.67        | 0.49        | 0.54        | 39.59    |
> | **2**    | 0.66        | 0.48        | 0.53        | 39.07    |
> | **3**    | 0.63        | 0.47        | 0.51        | 38.79    |
> | **4**    | 0.65        | 0.47        | 0.52        | 38.82    |
>
> **Decoding Consistency Across Seeds (Base Seed = 42)**
>
> **W-3:** Thank you for pointing this out. We agree that some methodological descriptions can be made clearer, and we will refine these parts in the revision. Regarding the initialization of the draft tree, the process is exactly as stated: we simply sample a small number of tokens randomly from the vocabulary. There is no additional mechanism or hidden assumption involved. We will make this explicit in Section 3.2 to avoid any ambiguity.

---

### Official Review · Reviewer_stwC · 2025-11-01

**Soundness:** 3
**Presentation:** 4
**Contribution:** 3
**Rating:** 6
**Confidence:** 3

**Summary:**

This paper proposes Progressive Tree Drafting (PTD), a training-free speculative decoding framework for LLM inference acceleration. Building on the insight that LLMs exhibit semantic robustness under controlled perturbations, PTD extends prior approaches like Self-Draft by organizing draft generation into a progressively expanded tree structure with prefix-sharing, branch pruning, and branch-wise perturbation. The method generates multiple coherent draft sequences and verifies them in parallel without requiring auxiliary draft models or architectural changes. Experiments across multiple LLaMA, Qwen, and CodeLLaMA models show up to 2× throughput improvement, outperforming LADE and Self-Draft, particularly on GSM-8k and MBPP, with competitive generation quality.

**Strengths:**

1. Method is well-motivated with illustrative diagrams (tree expansion, mask structure) and formal algorithm descriptions.

2. The experimental evaluation covers general QA, math reasoning, and coding tasks, with consistent speedups. The ablation studies on tree depth, branch width, and sampling strategies clearly demonstrate the characteristics of the proposed method.

3. The results provide stronger performance than both LADE and Self-Draft, highlighting real gains over recent state-of-the-art baselines.

**Weaknesses:**

1. Limited comparison to tree-based speculative frameworks (e.g., SpecInfer, EAGLE-2 dynamic draft trees).

2. The method still introduces non-trivial verification overhead, and overhead trends at larger scales (>32B parameters, >4K tokens) are not fully explored.

3. The choice of specific hyperparameters (e.g., max children=4, depth=6) appears arbitrary without an analysis of their impact across different model scales or task types.

**Questions:**

Please refer to Weaknesses.

---

> ### Author Response · Authors · 2025-11-23
> **Reply to Reviewer stwC**
>
> **W-1:** As summarized in **General Response 2,** existing tree-based frameworks differ from PTD in both drafting mechanism and execution pipeline, and therefore cannot be directly used as comparable baselines. We will clarify these distinctions in the revision.
>
> **W-2:** Thank you for raising this point. We agree that understanding performance under extremely long-context settings (e.g., >4K–8K tokens) can be valuable in general. However, the goal of PTD is to accelerate standard single-model speculative decoding under the widely adopted inference regime, where context lengths fall within the 1–4K range in mainstream applications such as chat, QA, math, and code generation.
>
> To further validate the scalability with respect to model size, we conducted additional experiments using a 70B model (Llama-3.3-70B-Instruct) under standard context settings. The results are summarized in the table below and demonstrate that PTD continues to provide consistent speedup at larger model scales.
>
> | Benchmark | AR          | PTD         |
> | --------- | ----------- | ----------- |
> | MT-Bench  | 5.51 ± 0.07 | 9.5 ± 1.37  |
> | GSM-100   | 5.55 ± 0.01 | 11.9 ± 2.21 |
> | Humaneval | 5.53 ± 0.02 | 10.8 ± 0.94 |
> | MBPP      | 5.55 ± 0.01 | 10.8 ± 1.13 |
>
> **The acceleration performance of PTD on large-scale language models(Llama-3.3-70B-Instruct)**
>
> **W-3:** Thank you for the comment. As detailed in **General Response 3**, the hyperparameters such as max children and depth primarily serve to control drafting overhead. Our analysis in Figure 5 shows that PTD is robust across a broad range of values, and the optimal settings depend on model size and hardware. We will clarify this more explicitly in the revision.

---

### Author Response · Authors · 2025-11-23
**General response**

### **1. On comparisons with other speculative decoding methods (EAGLE series, Medusa, SpecInfer, OPT-Tree, etc.)**

We noticed that several reviewers suggested comparing PTD with speculative decoding methods that require additional training or architectural modifications, including structure-altering approaches such as the EAGLE series and Medusa, as well as draft-model–based methods like SpecInfer and OPT-Tree. In the revision, we will further clarify the design goals of PTD and explain why we did not directly compare against these methods.

All of these approaches rely on **an additional draft model or architecture-level modifications**, and typically require model-specific training for each target model. Moreover, training configurations may need to be retuned across different hardware settings. Although the training cost for draft models is much lower than that of full LLMs, this requirement still reduces deployment flexibility. In addition, the extra draft model introduces additional computational and communication overhead during inference, which greatly limits its applicability in resource-constrained scenarios such as memory-constrained GPUs and edge/on-device settings.

In contrast, **PTD requires no additional training or auxiliary models**, and accelerates inference solely by reorganizing the decoding process itself. We acknowledge that certain advanced methods relying on extra models or training may achieve higher speedups, but PTD represents a **new, training-free direction for speculative decoding**, aimed at enhancing deployment flexibility and model-agnostic applicability, For example . Therefore, in this work we primarily compare PTD with the original single-model speculative decoding baseline to demonstrate its effectiveness, rather than with the latest multi-model or training-based frameworks.

------



### **2. On the relationship and differences between PTD’s tree structure and existing tree-based speculative decoding methods (EAGLE-2, SWIFT, Spec-LLaVA, SpecVLM, OPT-Tree, etc.)**

We appreciate the reviewers’ attention to this issue. Although PTD also adopts a tree structure, its **drafting and updating mechanisms** differ fundamentally from existing tree-based speculative decoding methods, which makes those methods incompatible with PTD’s framework.

In PTD, the tree structure is updated **in parallel and in sync** with the deocding (verification) process. The tree continuously expands, steps forward, and prunes throughout the entire decoding process, avoiding the sequential dependency of “draft first → verify later.” While PTD’s single-step expansion resembles the expansion strategies used in prior methods (and Figure 7 in our paper includes comparisons of different expansion strategies), expansion must occur at **every decoding step** in PTD. Therefore, we must strictly control the tree size to prevent drafting overhead from dominating the end-to-end latency. Additionally, to maintain semantic coherence across steps, we designed the **stepwise pruning algorithm** presented in the paper.

In contrast, existing tree-based approaches typically rely on an **external draft module** to independently generate a draft tree for the current decoded context at each stage. The expansion of the draft tree and the actual decoding process are sequential, rather than jointly updated. These methods do not need to maintain the tree across steps or handle semantic consistency between layers; tree size is usually controlled simply via a drafting budget.

Therefore, in terms of **drafting mechanism, execution strategy, and algorithmic goals**, PTD’s tree structure differs substantially from existing tree-based speculative decoding methods. We will clarify these distinctions more explicitly in the revised version to avoid potential misunderstanding.

------



### **3. On the choice of hyperparameters for the tree structure (maximum depth and number of child nodes)**

The maximum tree depth and the number of child nodes per node are primarily used to control the additional drafting overhead. As shown in Figure 5, both hyperparameters have relatively wide ranges of effective configurations, and the acceleration performance is not sensitive to these parameters. We did not include a more exhaustive analysis in the paper because we believe that the **optimal configuration depends heavily on the specific model size and hardware environment**, and under PTD’s training-free setting, the cost of tuning these parameters is extremely low (far lower than training a draft model). Therefore, an in-depth discussion of the “optimal” hyperparameters is not central to the main contributions of this paper.

In the next revision, we will explicitly emphasize this rationale and provide additional clarification to help readers better understand the practical impact of these hyperparameters.

---

### Note · Authors · 2026-01-13

**Comment:**

We would like to withdraw our paper. Based on the feedback and our internal review, we realized that the current version requires more comprehensive experiments and a more detailed analysis to meet the high standards of the conference. We appreciate the reviewers' constructive comments.

**Withdrawal Confirmation:**

I have read and agree with the venue's withdrawal policy on behalf of myself and my co-authors.